# Evaluation of Individual Plant Growth Estimation in an Intercropping Field with UAV Imagery

**Norazlida Jamil [1,2,*], Gert Kootstra [1] and Lammert Kooistra [3]**

1   Farm Technology, Department of Plant Sciences, Wageningen University and Research, Droevendaalsesteeg 1, 6708 PB Wageningen, The Netherlands; gert.kootstra@wur.nl
2   Faculty of Fisheries and Food Sciences, Universiti Malaysia Terengganu, Kuala Nerus 21030, Terengganu, Malaysia
3   Laboratory of Geo-Information Science and Remote Sensing, Wageningen University and Research, Droevendaalsesteeg 3, 6708 PB Wageningen, The Netherlands; lammert.kooistra@wur.nl
*   Correspondence: norazlida.jamil@wur.nl; Tel.: +31-317-48-29-80

**Abstract:** Agriculture practices in monocropping need to become more sustainable and one of the ways to achieve this is to reintroduce intercropping. However, quantitative data to evaluate plant growth in intercropping systems are still lacking. Unmanned aerial vehicles (UAV) have the potential to become a state-of-the-art technique for the automatic estimation of plant growth. Individual plant height is an important trait attribute for field investigation as it can be used to derive information on crop growth throughout the growing season. This study aimed to investigate the applicability of UAV-based RGB imagery combined with the structure from motion (SfM) method for estimating the individual plants height of cabbage, pumpkin, barley, and wheat in an intercropping field during a complete growing season under varying conditions. Additionally, the effect of different percentiles and buffer sizes on the relationship between UAV-estimated plant height and ground truth plant height was examined. A crop height model (CHM) was calculated as the difference between the digital surface model (DSM) and the digital terrain model (DTM). The results showed that the overall correlation coefficient ($R^2$) values of UAV-estimated and ground truth individual plant heights for cabbage, pumpkin, barley, and wheat were 0.86, 0.94, 0.36, and 0.49, respectively, with overall root mean square error (RMSE) values of 6.75 cm, 6.99 cm, 14.16 cm, and 22.04 cm, respectively. More detailed analysis was performed up to the individual plant level. This study suggests that UAV imagery can provide a reliable and automatic assessment of individual plant heights for cabbage and pumpkin plants in intercropping but cannot be considered yet as an alternative approach for barley and wheat.

**Keywords:** individual plant; plant height; unmanned aerial vehicle; RGB imaging; intercropping; time series



## 1. Introduction

Monocropping is a common practice in many agricultural production systems. It is the practice of growing one crop species at a given time in a large field. The resulting uniform growth allows large-scale mechanization, which greatly improves efficiency by maximizing output and minimizing labor [1]. However, monocropping practices have several known downsides, such as soil compaction, caused by the heavy machinery involved [2], easy spreading of diseases and pests, due to the abundance of host plants resulting in excessive use of pesticides [3], and an increasing need for fertilizers, due to disturbance of the ecosystem [4]. Due to increasing demand for sustainable and ecological farming, the limits of monocropping systems are approaching, and the interest in alternative cropping systems, such as intercropping, is increasing.

Intercropping involves two or more crop mixtures grown in the same space during a crop cycle [5]. It offers ecological mechanisms for suppressing weeds [6], controlling

pests and diseases [7], conserving soil resources [8], increasing yields [9], and using light, space, and water more efficiently than monocropping [10]. Despite the clear potential benefits, intercropping is currently marginally applied in practice. One reason is the increased need for labor to maintain the fields. Another reason is the limited knowledge about intercropping systems. There is a need for a better understanding of the plant interactions for different combination of species, cultivars, and field layouts [5]. This requires vast amounts of spatial–temporal data to study plant growth and development in the field at the level of individual plants, which can only be obtained by automated digital monitoring systems.

Different physiological plants traits, such as chlorophyl content, canopy temperature, and photosynthesis rate, and agronomic plant traits, such as plant height, leaf length, grain length, and plant diameter, are relevant in studying plant growth and development in the field [11]. Monitoring of individual plant height is one of the most important traits, as it provides information on plant growth [12], crop yield [13], crop biomass [14], and plant health throughout the growing season [15]. Traditionally, a ruler was used to manually measure the plant height of a small number of plants in the field [16]. This process, however, is labor-intensive and slow, resulting in only a limited sample size. Moreover, manual measurements are subjective and prone to errors [17]. Therefore, a fast, autonomous, and objective method to estimate plant height is needed to increase the quantity and quality of data.

Monitoring crops with cameras from unmanned aerial vehicles (UAV) has the potential to cover large areas with a high spatial resolution, making it an efficient method to estimate plant height [18]. With the rapid advancement of photogrammetric techniques, such as structure from motion (SfM) algorithms, UAVs mounted with high resolution RGB digital cameras can be used to make 3D reconstructions of the field [12]. Previous studies have used SfM on RGB images to obtain crop height for monocropping in sorghum [19], maize [20], and wheat [21]. Moreover, SfM was demonstrated to be able to derive plant height for various scale levels, such as orchard, field, plot, and patch levels [22,23]. Study [24] reported on the growth of Chinese cabbage and white radish with SfM on the plot level and [11] estimated the vegetable crop heights of eggplant, tomato, and cabbage with SfM on a patch level. Information about individual plant height, however, allows assessing trait variation and dynamic changes of plant interactions [25]. To our knowledge, only a few studies focus on plant height estimation at the individual plant level. Some studies dealt with monitoring of individual trees in mainly monocropping systems, such as citrus [22], palm [23], olive [26], cypress, cedar [27], and peach tree [28]; others have reported on the estimation of individual plant height of herbaceous species, such as artichoke [25]. None of these studies deal with individual plant height in intercropping systems. To fill the knowledge gap, the present study investigates an approach to estimate and monitor plant height of individual plants in an intercropping system using a UAV platform equipped with a RGB camera.

The overall goal of this study was to evaluate the accuracy of the estimation of individual plant height from UAV-based imagery for an intercropping field for four types of crops—barley, wheat, pumpkin, and cabbage—under varying conditions over the growing season. The specific objectives included the following: (1) to evaluate the estimated plant height from UAV-based imagery through comparison with ground truth plant height for the different crop species; (2) to compare different methods of estimating plant height from the UAV-based imagery and investigate differences between the four selected crops; (3) to evaluate the capability of monitoring plant growth over the growing season using a UAV-based approach.

## 2. Materials and Methods

This section provides an overview of materials and methods for this study. Section 2.1 gives a description of the experimental site. A description of the UAV-RGB image acquisition is given in Section 2.2. The ground truth measurements of individual plant height

are described in Section 2.3. Section 2.4 shows the steps for estimating of plant height from UAV imagery. The generation of the digital surface model (DSM) and the crop height model (CHM) are explained in Sections 2.4.1 and 2.4.2, respectively. Finally, Section 2.5 describes the statistical analysis used.

### 2.1. Experimental Sites

The field experiment was conducted from May to October 2020 on the Droevendaal organic experimental farm located in Wageningen, the Netherlands, at 51°59′28.4″ N 5°39′38.4″ E. A large-scale, long-term intercropping experiment at this location has been in place since 2018. The dimensions of the individual strips were 3 m by 54 m (Figure 1) which can be operated with current standard farm equipment. Wheat, barley, cabbage, potato, grass, and pumpkin were planted as crops in the experiment for this year. Four crops were selected for this study depending on the pairing of crops. Cabbage was grown in strips with neighboring strips of wheat, and pumpkin was neighboring with barley strips. The neighboring crops are part of a crop rotation commonly applied in organic cultivation systems in the Netherlands and were selected because of their ability to withstand competition and their support for natural enemies. All crops in the experiment were managed according to organic standards [29,30].

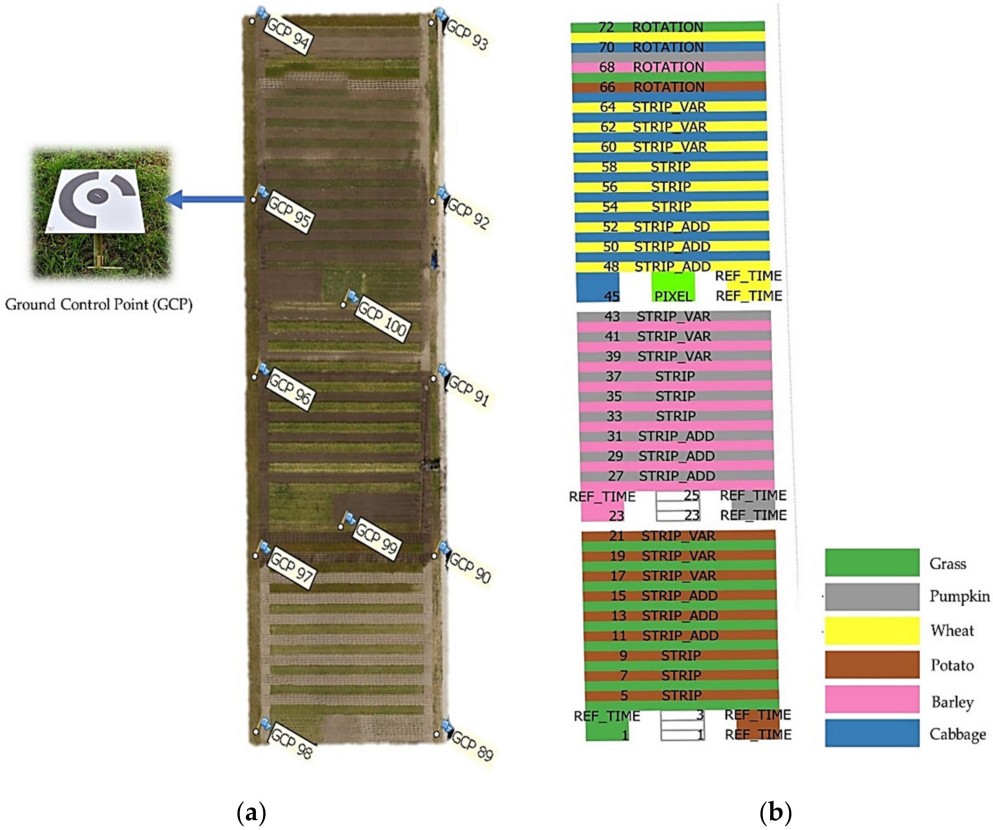

(**a**)                                                                 (**b**)

**Figure 1.** Field experiment overview for 2020; (**a**) strip cropping in the experimental strip cropping field at the Droevendaal organic experimental farm in Wageningen, the Netherlands—the location of the ground control points (GCP); (**b**) experimental design of the experimental strip cropping field. The labels in the strips refer to different strip cropping treatments: STRIP_VAR (mixed variety strips), STRIP_ADD (main crop and legume), and STRIP (sole crop strips). The strips for ROTATION, REF_TIME, and PIXEL were not included in this study.

A set of 12 ground control points (GCP) were strategically placed throughout the experimental field for geo-referencing of the UAV images (Figure 1a). A total of 10 GCPs (GCP 89–GCP 98) were placed around the field with 53 m distance and 2 GCPs (GCP99–

GCP 100) were placed in the middle of the field. The GCPs were made from aluminum sheets printed with markers (30 × 30 cm) (Figure 1). The location of the center point of the GCPs was measured with a real-time kinematic (RTK) positioning technique (GPS) to provide a sub-decimeter positioning accuracy of 2 cm and 5 cm in the horizontal and vertical directions, respectively.

### 2.2. UAV-RGB Image Acquisition

A total of 16 data collection campaigns were conducted during the growing season (Table 1). Wheat and barley were monitored from the early stem elongation stage until the late senescence stage. Meanwhile, cabbage and pumpkin crops were monitored and measured from the early seedling stage until the mature stage. The local weather research station (Meteobot) provided illumination and wind speed data. For every campaign, RGB images of the field were taken from 25 m height by a UAV DJI Matrice 210. The UAV was equipped with the Zenmuse X7 RGB camera, a high-performance RGB imaging system with a resolution of 6016 × 4008 pixels, and the DJI DL 50 mm F2.8 LS lens type. The flight time for the UAV was approximately 12 min per campaign to cover the whole area of the experimental field (Figure 1).

**Table 1.** UAV acquisition and ground truth measurement details observed for the experimental strip cropping field of the Droevendaal organic experimental farm in Wageningen, the Netherlands for 2020. The numbers in the first column refer to the number of field measurements per crop type.

| Crops (Number of Plants) | UAV Flight and Ground Measurement Dates | Flight Time | Illumination (lux) | Sky Condition | Wind (ms$^{-1}$) |
|---|---|---|---|---|---|
| Wheat (500); Barley (500) | 15 May 2020 | 13:21 | 328.3 | Sunny | 4.6 |
| Wheat (500); Barley (500) | 26 May 2020 | 14:00 | 708.3 | Sunny | 2.9 |
| Wheat (500); Barley (500); Cabbage (200) | 3 June 2020 | 13:32 | 643.3 | Sunny | 6.6 |
| Wheat (500); Barley (500); Cabbage (200) | 10 June 2020 | 12:37 | 165 | Cloudy | 3.6 |
| Wheat (500); Barley (500); Cabbage (193) | 17 June 2020 | 13:35 | 598.3 | Variable | 4.6 |
| Wheat (500); Barley (500); Cabbage (190) | 24 June 2020 | 13:14 | 746.7 | Variable | 7.1 |
| Wheat (500); Barley (500); Cabbage (189); Pumpkin (135) | 1 July 2020 | 16:07 | 348.3 | Cloudy | 10.1 |
| Wheat (500); Barley (500); Cabbage (187); (Pumpkin (137) | 8 July 2020 | 16:57 | 198.3 | Sunny | 2.6 |
| Wheat (500); Cabbage (187); (Pumpkin (136) | 15 July 2020 | 13:25 | 465 | Variable | 3.7 |
| Wheat (500); Cabbage (187); (Pumpkin (136) | 22 July 2020 | 13:48 | 680 | Cloudy | 2.3 |
| Cabbage (187); (Pumpkin (136) | 4 August 2020 | 14:45 | 255 | Variable | 1.4 |
| Cabbage (186); (Pumpkin (136) | 13 August 2020 | 13:38 | 516.7 | Variable | 2.5 |
| Cabbage (186); (Pumpkin (136) | 28 August 2020 | 11:59 | 265 | Sunny | 4.5 |
| Cabbage (185); Pumpkin (136) | 11 September 2020 | 12:55 | 485 | Sunny | 3.4 |
| Cabbage (185); Pumpkin (128) | 24 September 2020 | 11:28 | 403.3 | Sunny | 1.7 |
| Cabbage (185) | 7 October 2020 | 13:32 | 230 | Sunny | 9 |

### 2.3. Ground truth Measurements of Individual Plant Height

To evaluate the accuracy of height estimation using the UAV data, ground truth data were established around the same day as the UAV flight by manual, in-field height measurements. Plant height was defined as the distance from the ground to the highest point of the plant. The height was measured manually with a 5 m measuring tape. The measurements differed depending on the type of crop and plant growth stage, as shown in Figure 2.

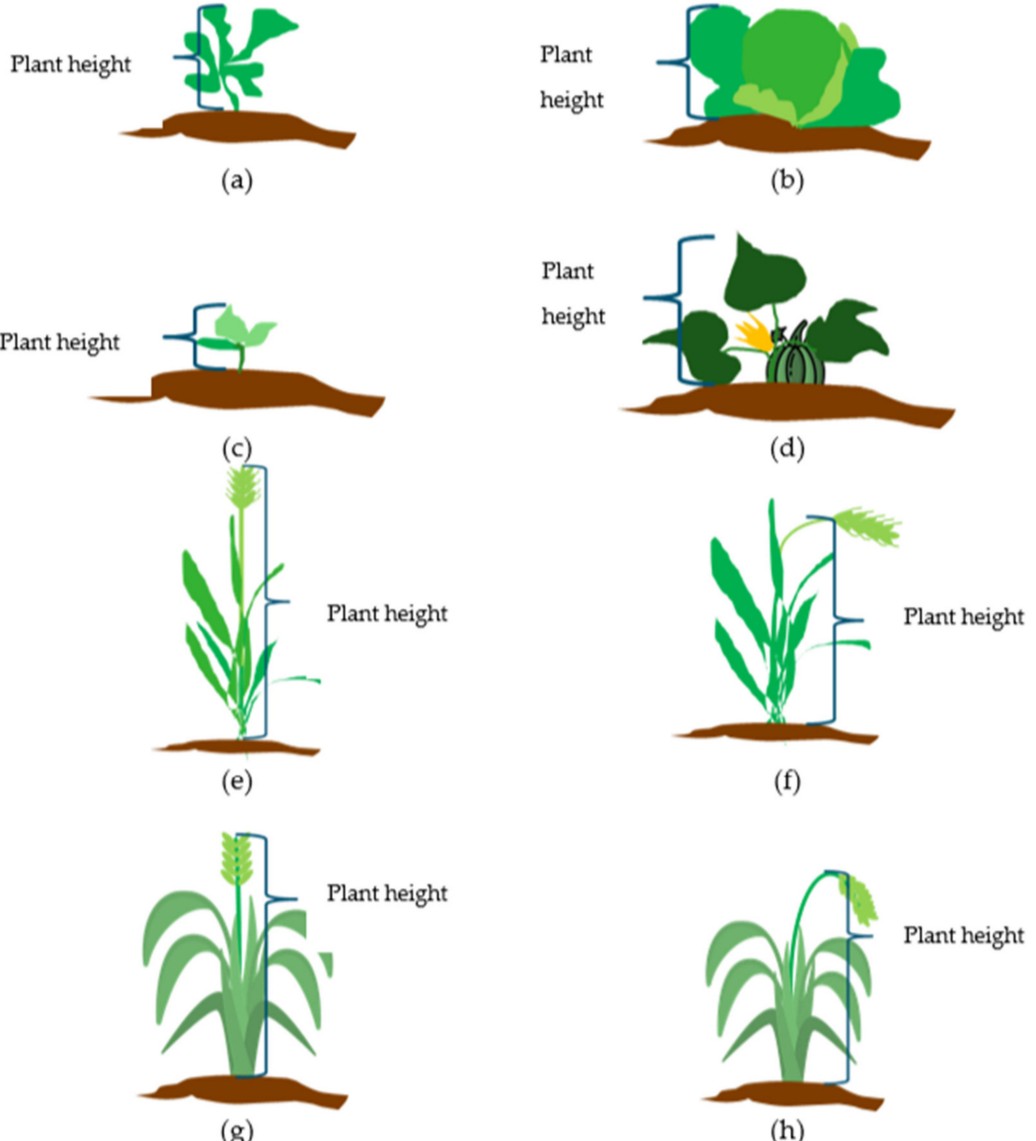

**Figure 2.** Plant height measurements according to growth stages for cabbage (**a**,**b**); pumpkin (**c**,**d**); barley (**e**,**f**); wheat (**g**,**h**).

For cabbage (Figure 2a,b) and pumpkin (Figure 2c,d), the ground truth plant height was measured from the base of the soil to the highest point of the individual plant during the complete growth period. For the cereal crops (barley and wheat), in the vegetative stage, the height was taken from the soil surface vertically to the top of the spike (Figure 2e,g). During the late ripening stage, height was measured from the ground to the neck of the spike (Figure 2f,h).

For cabbage and pumpkin, 20 plants per strip were identified for the manual measurements. For cabbage, 200 plants were measured (10 strips, 20 plants per strip). Since several

cabbage plants died or were removed during the season, 185 plants were measured over the complete period. Initially, there were 180 pumpkin plants assigned to be measured (20 plants per strip, 9 strips); however, just before the first measurement day, the pumpkin strips were infected by blight disease that caused some of the plants to dry out, which lead to a final total of 128 plants being measured over the complete period. For the cereal crops, individual plants are too small to identify reliably in both the field as well as in the image data. We therefore used a small area of 50 × 50 cm in which we measured 5 individual plants, 1 in the center and 1 in each corner of the area. For both barley and wheat, 100 areas were measured (10 areas per strip, 10 strips), with a total of 500 plant height measurements. The individual plant height measured in the 50 × 50 cm area was averaged to obtain only one value per area, to be compared with the plant height extracted from the UAV over the same area.

Human error was introduced for barley and wheat height measurements in the middle of data collection due to the plant senescence at the last stage of plants development in the growing period. Thus, from our observations, the ground truth measurement for barley and wheat on 17 June 2020 and 24 June 2020 were recorded by bending up the spike instead of measuring from the ground level to the neck of the spike. Moreover, the individual plant locations were measured by RTK–GPS to be used to estimate the plant height.

The ground truth measurements required two researchers working for about 20 min per strip. The total time for the whole field to perform manual measurements was approximately 10 h for all 4 selected crops, depending on the weather conditions.

### 2.4. Overview of the Methodology Estimation of Plant Height from UAV Imagery

In this section, an overview of the analysis workflow is described using a flowchart (Figure 3) that summarizes how to calculate the individual plant height from the RGB images acquired by UAV. The steps to generate the digital surface model (DSM) are explained in Section 2.4.1. Section 2.4.2 explains how the DSM and digital terrain model (DTM) are used to calculate the plant height.

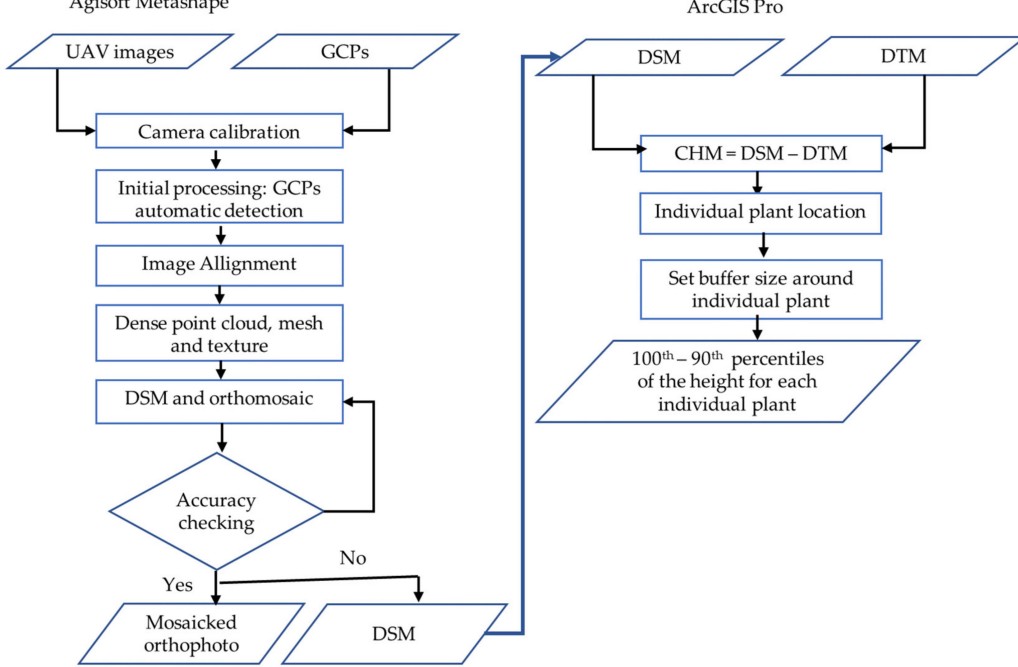

**Figure 3.** Flowchart depicting the general overview of the methodology for calculating the canopy height model (CHM) and the height of individual plants that involved processing in Agisoft Metashape and ArcGIS Pro software.

### 2.4.1. Generation of the Digital Surface Model

The Agisoft Metashape software (ver. 1.5.1, Agisoft LLC, St. Petersburg, Russia) was used to perform image alignment and to construct 3D point clouds, orthomosaics, and DSMs from the images taken by the UAV and the geographical coordinates of the 12 GCPs. The parameter settings used for Agisoft Metashape software are summarized in Table 2 and were applied for all the acquisition dates.

RGB image files were loaded into Agisoft Metashape, together with the locations of the GCPs measured by RTK–GPS. This software is a structure from motion with multi-view stereo (SfM-MVS)-based method, which estimates the camera poses and camera distortion parameters. Subsequently, this method creates a 3D reconstruction of the environment observable in the images. In this research, the focus lies on SfM-MVS, because the method does not need to be in real time and the data is already gathered; in addition, images for all data campaigns were available at the start of the process. For each flight campaign, the location of all the GCPs was measured and the average value was used for all dates. Then, the GCPs were automatically detected in the process and both camera and GCPs were used in the alignment step for reference. In this way, possible non-linear deformations resulting from the SfM-MVS process were removed by fitting the estimated point cloud on the known reference coordinates. This optimization adjusted the estimated point coordinates and camera parameters, effectively minimizing the sum of reprojection error and reference coordinate misalignment error. Rolling shutter compensation was enabled during the process because the Zenmuse X7 camera uses a rolling electronic shutter. The accuracy for aligning photos was set at the highest level and the quality in building the point cloud was set at ultra-high to achieve the best accuracy for the output data set.

At the dense point cloud stage, mild depth filtering was used to smooth the point cloud, which is important to achieve good plant height accuracy. After the geo-referencing process, from Agisoft Metashape, the following outputs were created: a 3D mesh and a digital surface model (DSM) from the dense point cloud. To simplify for further analyses, the DSMs were exported as GeoTiff-formatted image files.

### 2.4.2. Crop Height Model (CHM) Generation

The UAV-estimated individual plant height was calculated using ArcGIS Pro (version 2.8.0, Esri Inc., Redlands, CA, USA). In the first step, the DSMs generated in Agisoft Metashape were loaded into ArcGIS Pro. The digital terrain model (DTM) represents the field topography with no crop grown on the ground and the DSM is the state of a field with both crop features and underlying topography [31]. The Droevendaal organic experimental farm is an intercropping field, so it is difficult to obtain a DSM dataset for which all vegetation is removed at the beginning of the growing season. In this study, we used different DTMs for different crops accordingly. We choose the DSM of one day that has bare soil on the strips for each crop as our alternative reference DTM. For cabbage, pumpkin, barley, and wheat, the following dates were used to derive the DTM surface: 15 May 2020, 17 June 2020, 13 August 2020, and 28 August 2020, respectively. Then, the crop height models (CHMs) were generated by subtracting DSMs from the DTMs (Figure 4), as follows:

$$CHM = DSM - DTM \qquad (1)$$

The individual plant locations measured by RTK–GPS were used to determine the location of the plants for which the plant height was estimated. The buffer tool was used to create regions of interest (ROI) to extract pixels from the CHM and UAV-based individual plant height from each plant location. A circular buffer with certain radians were tested according to the crop type: 3 cm, 5 cm, and 10 cm for pumpkin and cabbage crops, and 50 cm and 75 cm for wheat and barley crops. The size chosen depends on the plant shape and size. Then, plant height values were extracted with 100th–90th percentiles from each buffer.

**Table 2.** Parameters for UAV image processing by Agisoft Metashape software.

| Process | Parameter | Setting |
|---|---|---|
| Reference setting | Coordinate system | Amersfoort/RD New (EPSG:28992) |
| | Camera reference | WGS 84 (EPSG:4326) |
| | Marker reference | Amersfoort/RD New (EPSG:28992) |
| | Camera accuracy (m) | 0.05 |
| | Camera accuracy (deg) | 10 |
| | Marker accuracy | 0.005 |
| | Scale bar accuracy | 0.001 |
| | Capture distance (m) | 20 |
| Detect GCPs | Number of GCPs | 12 |
| | Marker type | Circular 12 bit |
| | Tolerance | 70 |
| Camera calibration | Enable rolling shutter compensation | Yes |
| Align photos | Accuracy | Highest |
| | Generic preselection | No |
| | Reference preselection | No |
| | Key point limit | 40,000 |
| | Adaptive camera model fitting | Yes |
| Build point clouds | Quality | Ultra High |
| | Depth filtering | Mild |
| Build mesh | Source data | Dense cloud |
| | Surface type | Height field (2.5D) |
| | Face count | Medium |
| | Interpolation | Enabled |
| | Calculate vertex colors | Yes |
| Build texture | Mapping mode | Adaptive orthophoto |
| | Blending mode | Mosaic |
| | Enable hole filling | Yes |
| | Enable ghosting filter | Yes |
| Build DSM | Projection type | Geographic; Amersfoort/RD New (EPSG:28992) |
| | Source data | Dense cloud |
| | Interpolation | Enabled |
| Orthomosaic | Projection type | Geographic; Amersfoort/RD New (EPSG:28992) |
| | Surface | DSM |
| | Blending mode | Mosaic |

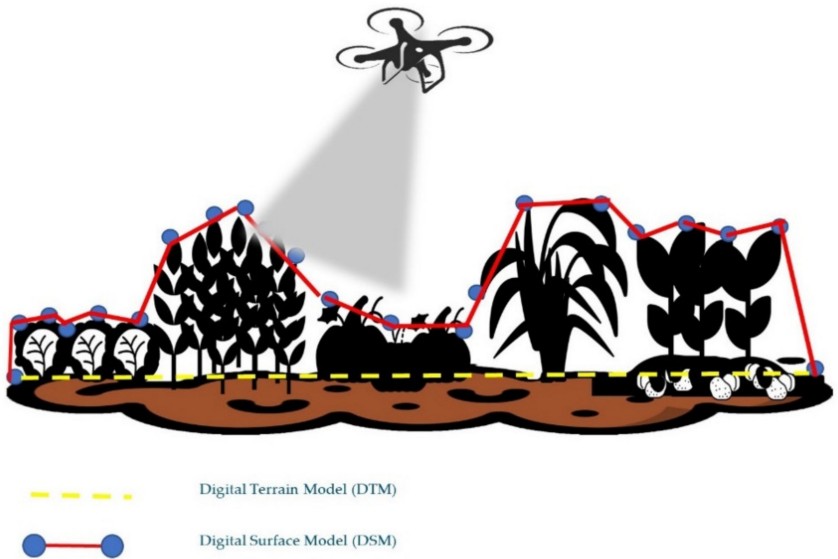

Digital Terrain Model (DTM)

Digital Surface Model (DSM)

**Figure 4.** Canopy height model (CHM) extracted from difference between DSM and DTM. The DTM needs to be derived for the field without crops.

### 2.5. Statistical Analysis

The accuracy of the estimation of plant height from the UAV data was analyzed using the ground truth measurements. For the accuracy of the individual plant height, 3 metrics were used—coefficient of determination ($R^2$), mean absolute error (MAE), and root mean square error (RMSE)—for different buffer sizes (3 cm, 5 cm, 10 cm, 50 cm, and 75 cm) and the 100th–90th percentiles (with steps of 1) were used to evaluate the strength of the relationship between plant heights, as in the following equations:

$$R^2 = \frac{\sum_{i=1}^{n}(y_i - \hat{y}_i)^2}{\sum_{i=1}^{n}(y_i - \overline{y}_i)^2} \tag{2}$$

$$MAE = \frac{1}{n}\sum_{i=1}^{n}|y_i - \hat{y}_i| \tag{3}$$

$$RMSE = \sqrt{\sum_{i=1}^{n}\frac{(\hat{y}_i - y_i)^2}{n}} \tag{4}$$

where n is the number of samples (individual plant) in the data set, $y_i$ is the measured ground truth height of plant, $i$, $y_i$ is the estimated plant height from the UAV data, and $\overline{y}_i$ indicates the average ground truth plant height.

To access the accuracy of estimation of the growth curves of individual plants from time 1 to time t, we calculated the correlation between the true growth curve $Y = \{y_i^1, y_i^2, \ldots y_i^t\}$ and the estimated $\hat{Y} = \{\hat{y}_i^1, \hat{y}_i^2, \ldots \hat{y}_i^t\}$: $r = \text{corr}(Y, \hat{Y})$.

## 3. Results

### 3.1. Development of Crop Height Models (CHMs) for Individual Plant Height

Figure 5 illustrates the condition of the experimental field at two points in time for cabbage, pumpkin, barley, and wheat crops, presenting the orthomosaics as well as the CHMs. The first row in Figure 5 shows an overview RGB image of the plants in the strips (rectangular shape). A small area (white rectangle) is zoomed in on the second row. In the second row, individual plant locations are indicated with the circles, which depict the buffer size used in the experiments. The third and fourth rows show the crop height models for the same regions, as depicted in the color images in the first two rows.

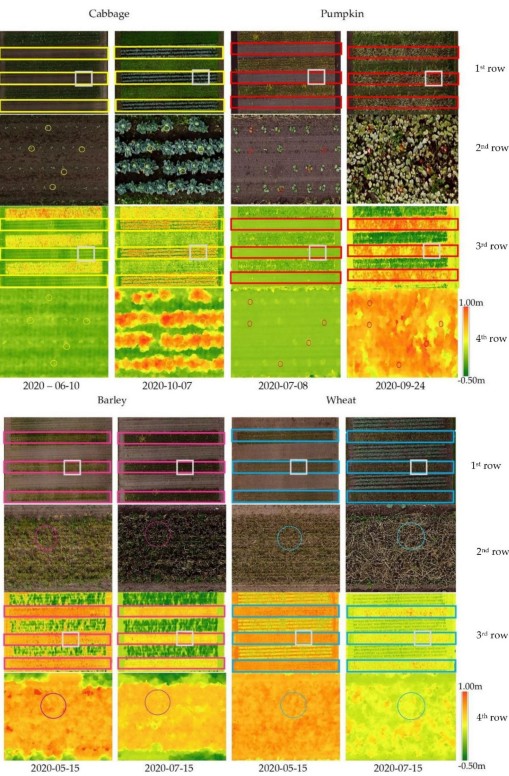

**Figure 5.** Visualization of the UAV-derived development of crop height in meters, as shown in orthomosaics and crop height models (CHMs) at the beginning and the end of the growing season. The CHMs were computed by subtracting DSM from DTM. A part of the orthomosaics and CHMs zoomed into one of the strips showing both the area with and without buffer. The colors in CHMs correspond to the value of the crop height.

The CHMs shows clear changes in height at different growing stages with dark green to light green color representing lower terrain elevation and yellow to red color representing the higher elevations. The difference in the elevation of the plants compared with the soil was relatively low at the seedling stage for cabbage and pumpkin crop but differences were considerable at the later stages of crop development. For barley and wheat, the crop height in the early stage was already considerable, because the first acquisition was performed at the crop stem elongation stage, where the nodes development thickens, and more leaf is produced. However, the CHMs for barley and wheat exhibit lower values at later stage due to plant senescence.

### 3.2. Optimal Setting for Plant Height Assessment in an Intercropping Field

To derive plant height from the CHMs, the optimal setting for the buffer area around the measurement point and the distribution threshold (the cut-off percentile) for the selected pixels in the buffer needed to be made. Figure 6 shows the average mean absolute error (MAE) on the left and the average root mean square error (RMSE) on the right for the 100th–90th percentile as distribution threshold for all crops throughout the growing season.

For cabbage and pumpkin crops, the average MAE and average RMSE values were calculated using a buffer size of 3 cm, 5 cm, and 10 cm, while for barley and wheat, a buffer size of 50 cm and 75 cm was used. An increase in the buffer size resulted in decreasing values of average MAE and average RMSE, indicating that a better height estimate can be made with a larger buffer size, increasing the likelihood that the highest point of the plant is in the buffer. The error in height estimation decreased when a larger percentile was used, which suggests that the amount of noise in the CHM is low, and the maximum height value in the buffer can be taken to estimate the plant height. The optimal settings for cabbage

and pumpkin were achieved with the 100th percentile as the distribution threshold and a 10 cm buffer size.

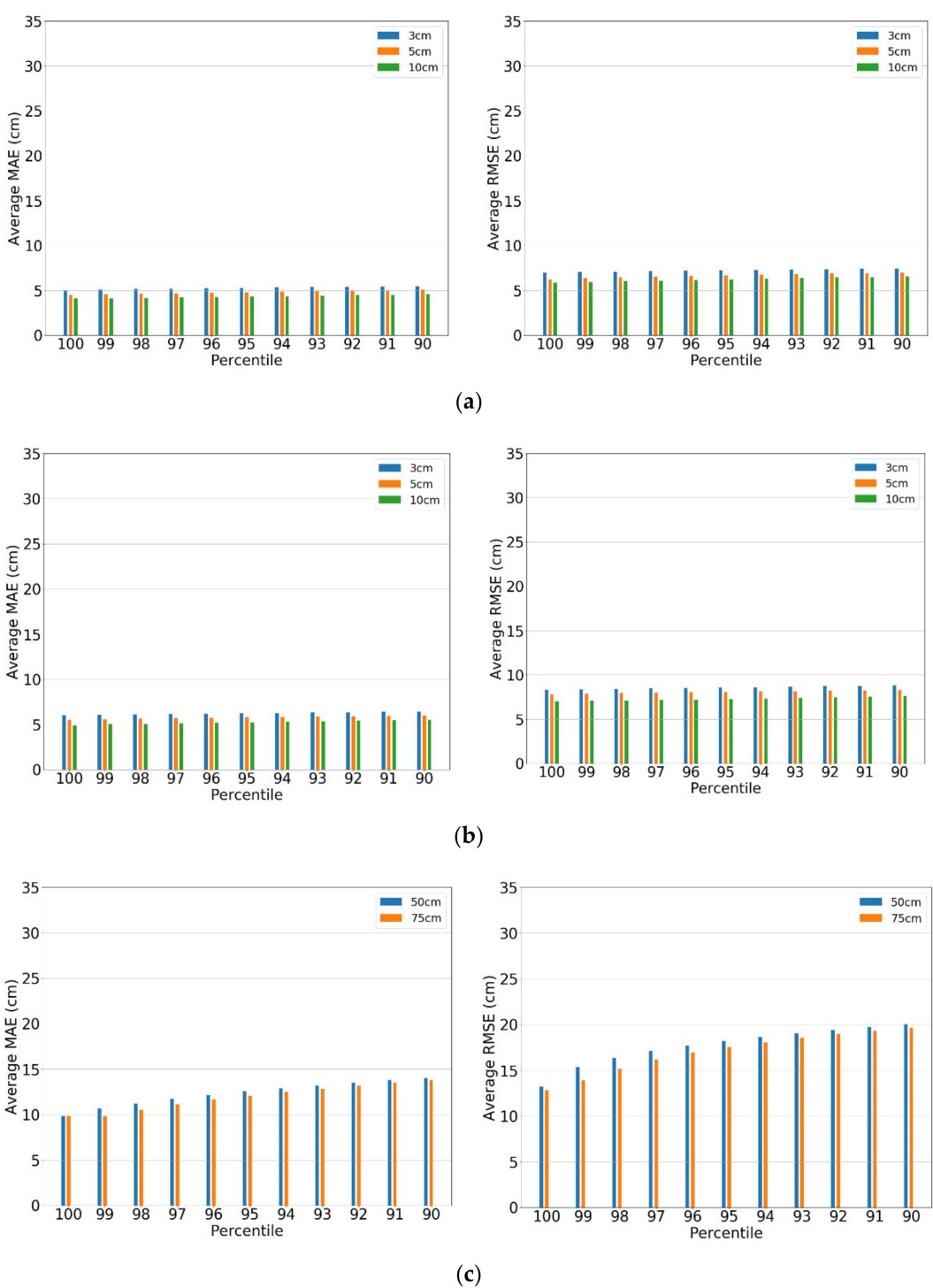

**Figure 6.** *Cont.*

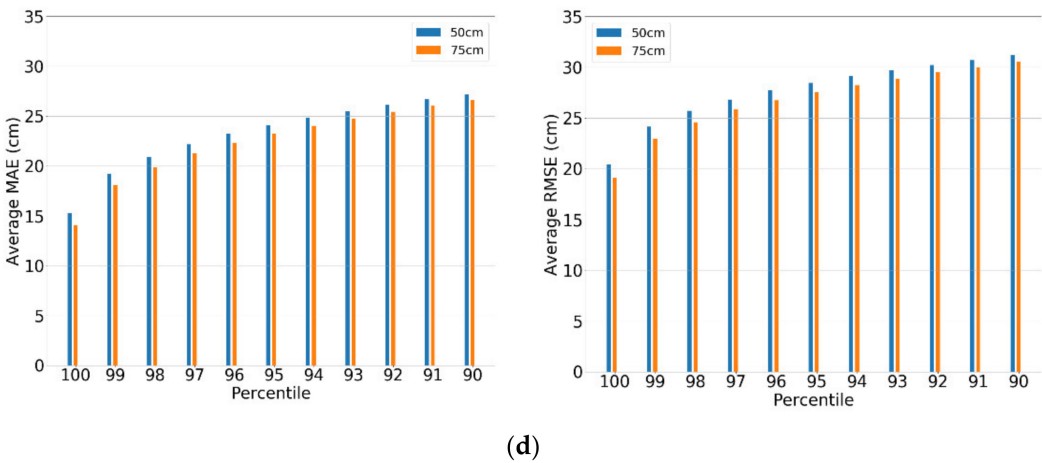

(**d**)

**Figure 6.** Bar plot of average MAE (cm) and average RMSE with the 100th–90th percentile of (**a**) cabbage; (**b**) pumpkin; (**c**) barley; and (**d**) wheat, with different buffer size used in the intercropping experimental field over growing season.

The results in Figure 6 for wheat and barley crop also show that a larger buffer size and a larger percentile improve the height estimation. The optimal performance was achieved with the 100th percentile and 75 cm buffer size. However, the 99th percentile was chosen for further analysis because it showed to be more consistent for plant height estimation at different growth stages, especially during the senescence stage. The decision was influenced by the observations of the presence of weeds on the latest date which were sometimes higher than the barley and wheat plants.

The final decision for the optimal settings that were used for further analysis is shown in Table 3.

**Table 3.** Optimal parameter setting for the UAV-based height estimation consisting of percentile and buffer size for cabbage, pumpkin, barley, and wheat crops for an intercropping field.

| Parameter | Cabbage | Pumpkin | Barley | Wheat |
|-----------|---------|---------|--------|-------|
| Percentile | 100th | 100th | 99th | 99th |
| Buffer size | 10 cm | 10 cm | 75 cm | 75 cm |

Table 4 shows the $R^2$, MAE, and RMSE results for the plant height estimation for the different crops on all dates using the optimal setting according to the crop types. Both cabbage and pumpkin crops have high $R^2$ values for all observations over the whole season of 0.8601 and 0.9366, respectively, and the MAE and RMSE values are low, with 4.78 cm and 6.75 cm for cabbage, and 4.91 cm and 7.00 cm for pumpkin. The $R^2$ values for the individual dates are lower, indicating that it is more difficult to explain the variance on the individual dates. The highest $R^2$ for cabbage was 0.5426 on 22 July 2020 with a MAE value of 3.98 cm and RMSE of 4.49 cm at the early heading of the crop stage. Most of the results obtained for pumpkin crop had higher $R^2$ values and lower MAEs and RMSEs at the earlier data campaign compared with those at the later date with the highest $R^2$ value of 0.5936, MAE value of 1.99 cm, and RMSE of 2.53 cm.

**Table 4.** Correlation ($R^2$), mean absolute error (MAE), and root mean square error (RMSE) for the relation between UAV-derived height estimation and field measurements for cabbage, pumpkin, barley, and wheat crops for all flight campaigns. Buffer sizes: cabbage—10 cm; pumpkin—10 cm; barley—75 cm; wheat—75 cm. Percentiles: cabbage—100th; pumpkin—100th; barley—99th; wheat—99th.

| Date | Correlation, $R^2$ | | | | Mean Absolute Error (cm) | | | | Root Mean Square Error (cm) | | | |
|---|---|---|---|---|---|---|---|---|---|---|---|---|
| | Cabbage (10 cm) | Pumpkin (10 cm) | Barley (75 cm) | Wheat (75 cm) | Cabbage (10 cm) | Pumpkin (10 cm) | Barley (75 cm) | Wheat (75 cm) | Cabbage (10 cm) | Pumpkin (10 cm) | Barley (75 cm) | Wheat (75 cm) |
| 15 May 2020 | - | - | 0.3817 | 0.1229 | - | - | 22.0601 | 26.2932 | - | - | 23.6216 | 26.5278 |
| 26 May 2020 | - | - | 0.4080 | −0.0647 | - | - | 22.0794 | 35.0511 | - | - | 26.0833 | 35.8434 |
| 3 June 2020 | 0.0522 | - | 0.7242 | 0.5388 | 14.0368 | - | 8.2577 | 29.3145 | 14.6421 | - | 13.6089 | 30.5235 |
| 10 June 2020 | 0.0405 | - | 0.7471 | 0.4782 | 3.4587 | - | 3.3903 | 8.9060 | 5.0433 | - | 4.2933 | 10.7075 |
| 17 June 2020 | 0.1513 | - | 0.1521 | 0.0947 | 4.1423 | - | 5.6804 | 18.1641 | 4.8292 | - | 7.5138 | 19.4909 |
| 24 June 2020 | 0.2246 | - | 0.0295 | 0.0776 | 8.3127 | - | 8.0717 | 26.6474 | 9.5656 | - | 10.0884 | 29.0250 |
| 1 July 2020 | 0.3107 | 0.3145 | 0.1917 | 0.1844 | 3.0811 | 2.4888 | 5.2924 | 3.9100 | 3.9869 | 3.3732 | 7.0411 | 5.0911 |
| 8 July 2020 | 0.3954 | 0.5963 | −0.0060 | 0.2404 | 3.1774 | 1.9909 | 6.5550 | 5.8881 | 4.3547 | 2.5253 | 7.6256 | 7.7938 |
| 15 July 2020 | 0.4698 | 0.5227 | 0.1151 | 0.1865 | 4.7596 | 3.2173 | 7.8664 | 10.2552 | 5.9438 | 4.3638 | 9.8515 | 12.3438 |
| 22 July 2020 | 0.5426 | 0.3336 | - | 0.0754 | 3.9839 | 4.7967 | - | 16.7831 | 4.4981 | 6.5801 | - | 19.0083 |
| 4 August 2020 | 0.3008 | 0.2925 | - | - | 2.9135 | 6.6935 | - | - | 5.0297 | 8.6732 | - | - |
| 13 August 2020 | 0.1219 | 0.1695 | - | - | 4.7442 | 6.1096 | - | - | 7.2952 | 7.8323 | - | - |
| 28 August 2020 | 0.1850 | 0.2915 | - | - | 3.4350 | 6.8698 | - | - | 5.4805 | 8.7165 | - | - |
| 11 September 2020 | 0.3365 | 0.4641 | - | - | 3.5655 | 5.9139 | - | - | 4.8436 | 7.6058 | - | - |
| 24 September 2020 | 0.3714 | 0.5382 | - | - | 3.1677 | 6.2345 | - | - | 4.3128 | 9.6348 | - | - |
| 7 October 2020 | 0.3527 | - | - | - | 3.4409 | - | - | - | 4.6963 | - | - | - |
| $R^2$ values for all dates (all CHMs) | 0.8601 | 0.9366 | 0.3563 | 0.4949 | 4.7772 | 4.9149 | 9.9170 | 18.1209 | 6.7547 | 6.9977 | 14.1616 | 22.0398 |
| $R^2$ values for all dates (without two earliest CHMs) | - | - | 0.3635 | 0.1973 | - | - | 6.4445 | 14.9832 | - | - | 8.9972 | 18.9375 |

The whole season $R^2$ values for barley and wheat crops were lower at 0.3563 and 0.4949, respectively, with MAE and RMSE of 9.92 cm and 14.16 cm for barley, respectively, and 18.12 cm and 22.04 cm for wheat, respectively. For the individual days, a large variation in the accuracy was observed. The highest $R^2$ values for barley and wheat crops were obtained at the stem elongation stage with $R^2$ of 0.7471 and 0.4782, respectively, both for the date of 10 June 2020.

The error in height estimations for the first two dates for barley were particularly high, possibly due to errors in the SfM-MVS process. When the data at the earliest two dates were removed, the overall $R^2$ for barley slightly increased to 0.3635 and the overall MAE and overall RMSE were reduced to 6.44 cm and 8.99 cm, respectively. However, when the same approach was applied to the wheat crop, overall $R^2$ values were expected to increase also; however, instead, a lower $R^2$ value was obtained. The overall MAE and RMSE did improve when compared with data including all the dates of measurement.

Scatter plots showing the relationship between the UAV-based plant height estimations and the manual measurements are depicted in Figure 7 for each crop for all dates. Each dot on the scatter plots represents an individual plant. The confidence ellipses in the scatter plots show varying strengths of the correlation between ground truth and UAV-estimated plant height. For cabbage (Figure 7a), the values were scattered quite far from the straight line at the seedling stages and scattered close to the diagonal line at the later growth stages,

which indicates a better accuracy of the UAV-based estimations at the later growth stages. For pumpkin (Figure 7b), the points are generally close to the ideal line, with a slight underestimation at earlier dates. At later dates, the points are scattered around the line, but with a larger spread. The confidence ellipses are generally oriented diagonally, indicating a correlation between the ground truth and UAV-estimated plant height.

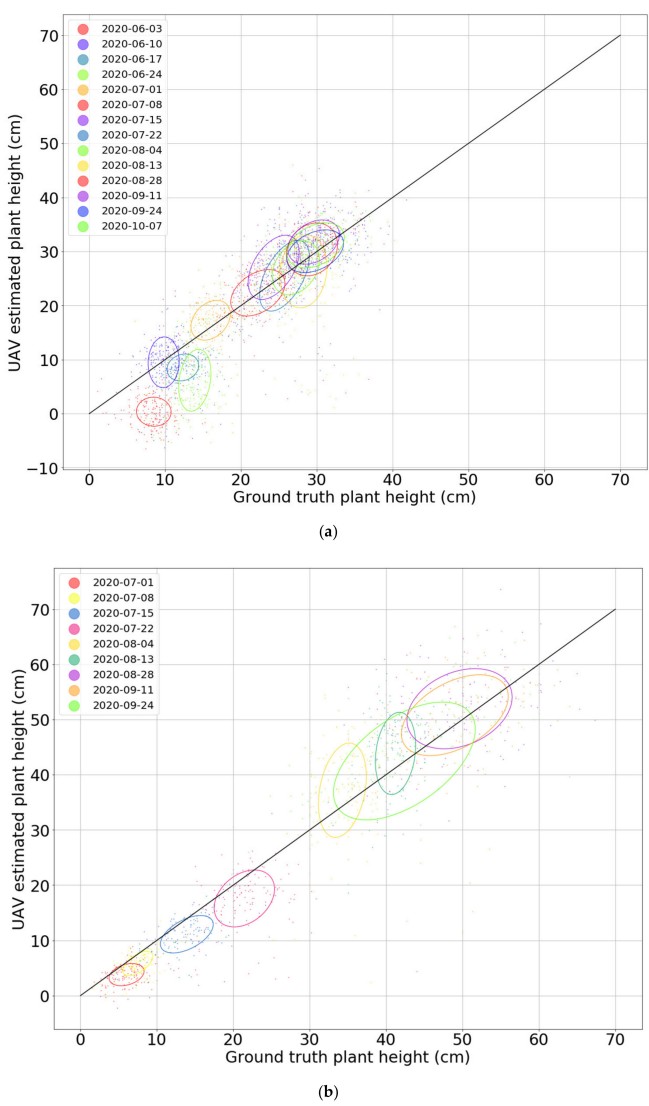

**Figure 7.** *Cont.*

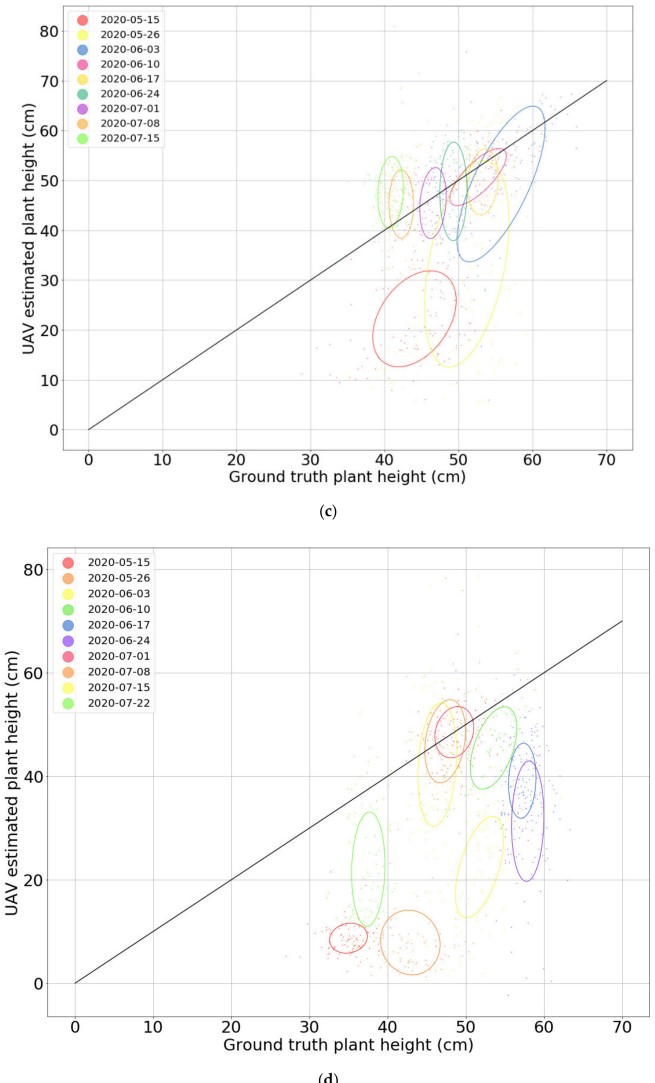

**Figure 7.** Scatter plots of the UAV-estimated plant height compared with the ground-truth plant height for (**a**) cabbage, (**b**) pumpkin, (**c**) barley, and (**d**) wheat. The plots include the 95% confidence ellipses of the covariance matrices per date.

The confidence ellipses in Figure 7c,d for barley and wheat have more circular shape and are sometimes even oriented vertically, showing a low correlation between UAV-estimated and ground truth plant height. For a few dates sever underestimations are visible, which are caused by a poor 3D reconstruction of the canopy height by the SfM-MVS method.

### 3.3. Individual Plant Growth

Here, growth curves for individual plants estimated by the UAV data are compared with the growth curves generated from the manual measurements. Figure 8 shows box and whisker plots for the correlation coefficients comparing the UAV-based with the manual measurements for each crop. For cabbage, the average correlation coefficient is 0.90 with a small range. For pumpkin, the average correlation coefficient is 0.96, also with a small range. On the other hand, the average correlation coefficients for barley and wheat are 0.32 and 0.52, respectively, and for both crops, a relatively large range is observed. The results show that monitoring of growth of individual plants can be performed for cabbage and pumpkin quite well. However, for barley and wheat, the correlations are relatively low and plant growth cannot be reliably monitored from the UAV data.

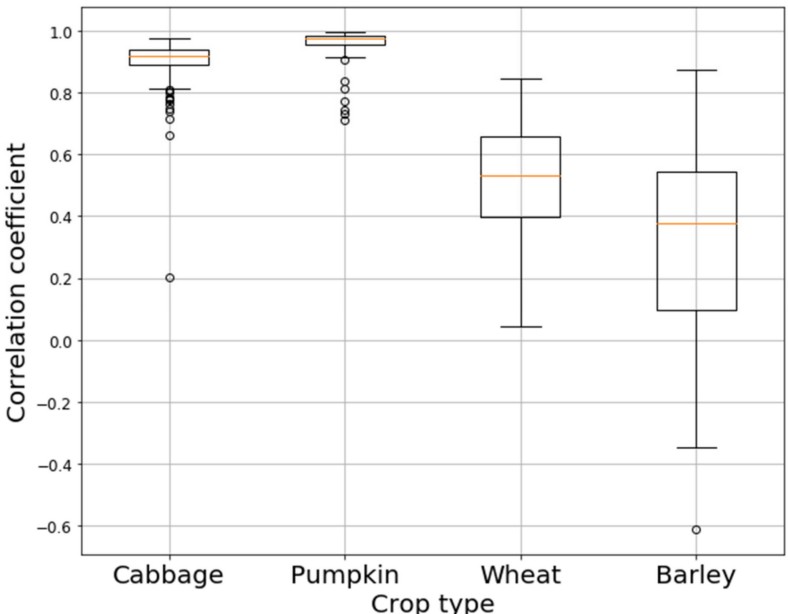

**Figure 8.** Boxplot of the correlation coefficient for the growth curves of individual plants generated from the UAV-estimated height and compared with the growth curves generated from the manual measurements over the complete growing season for cabbage, pumpkin, wheat, and barley plants.

Figure 9 shows a comparison between the ground truth growth curves based on the UAV data (solid lines) and based on the manual measurements (dashed lines) over the growing season. On the left (Figure 9a,c,e,g), two plants for each crop are shown, that have the highest correlation coefficient. On the right (Figure 9b,d,f,h), the two plants with the worst correlation coefficient are shown per crop.

Figure 9a shows that CB176 and CB191 have a high correlation between UAV-estimated height and ground truth height with correlation coefficient of 0.98 and 0.97, respectively. Figure 9b shows the two worst correlations for cabbage. The low correlation coefficient of 0.20 for CB084 can be explained by errors in the ground truth measurement. The UAV-estimated height of CB084 shows a decline of height trend on 8 July 2020 because CB084 has been removed from the experimental field; however, the height increased at the later stage because weeds grew on the empty space.

Figure 9c shows a very strong correlation for two pumpkin plants PK001 and PK008 with a 1.00 correlation coefficient. Even the two worst correlating plants, PK088 and PK063 in Figure 9d, still show decent correlation coefficients of 0.71 and 0.73, respectively. Here, the UAV-based estimations mainly go wrong on the last two dates.

The barley plants BA045 and BA190 in Figure 9e show a good correlation with correlation coefficients of 0.80 and 0.87, respectively. On the other hand, BA080 and BA310 in Figure 9f have negative correlation coefficients of $-0.61$ and $-0.35$, respectively.

For wheat, WH410 and WH245 shows good correlation coefficients of 0.83 and 0.84, respectively, in Figure 9g. Figure 9h shows no correlation between UAV-based and manual measurements for WH205 and WH270, with correlation coefficients of 0.04 and 0.05, respectively.

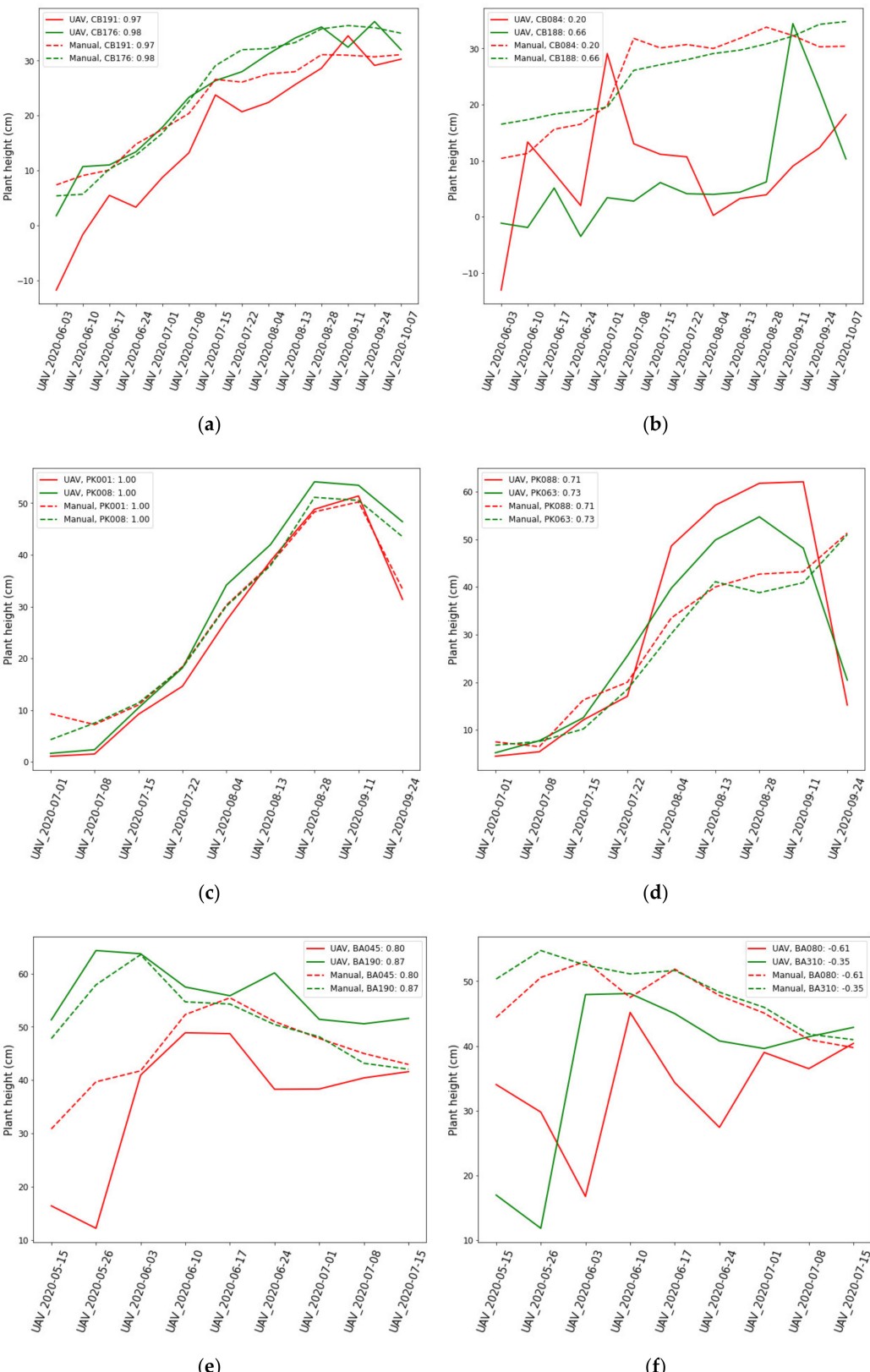

**Figure 9.** *Cont*.

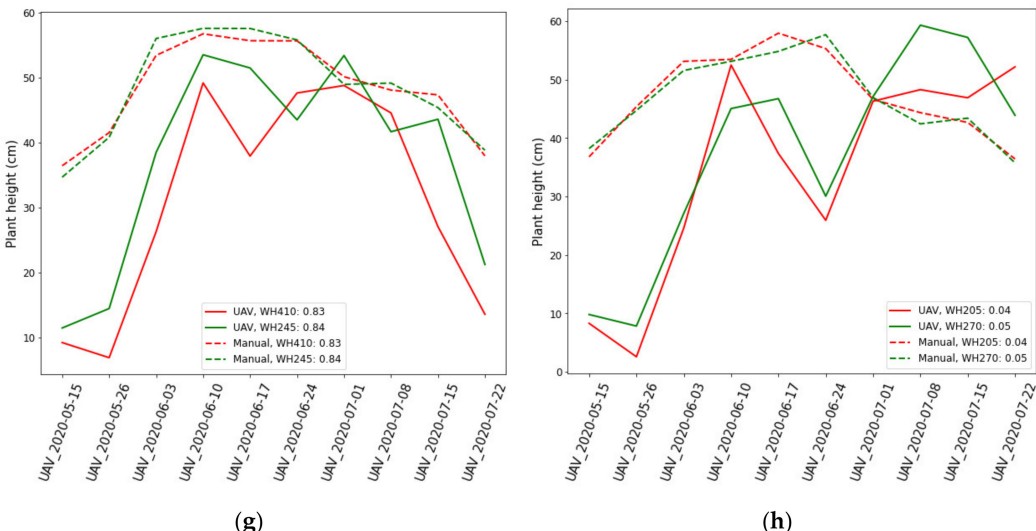

(**g**)                                                                  (**h**)

**Figure 9.** UAV-estimated individual plant growth and ground truth plant growth for two plants with a high correlation: (**a**) cabbage, (**c**) pumpkin, (**e**) barley, and (**g**) wheat. Additionally, for two plants with a low correlation: (**b**) cabbage, (**d**) pumpkin, (**f**) barley, and (**h**) wheat.

## 4. Discussion

### 4.1. Significant Results of Crop Height Models (CHMs) for Individual Plant Height

The goal of this study was to evaluate the accuracy of UAV-based estimation of plant height for four crops in an intercropping experiment (cabbage, pumpkin, barley, and wheat). After optimizing the buffer size and percentile, the results showed strong correlations with the ground truth data for cabbage and pumpkin with $R^2$ = 0.8601 and 0.9366 over the growing season, respectively. The average RMSE for these crops was 6.75 cm for cabbage and 6.99 cm for pumpkin (Table 4). Comparing the growth curves of individual plants estimated from the UAV imagery to the ground truth curves resulted in high correlations for these crops, with correlation coefficients of 0.90 and 0.96, respectively (Figure 8). Similar findings were reported in previous studies of monocultural cropping systems. Accuracies of crop height estimation for cabbage at 5 dates throughout the growing season were reported with $R^2$ of 0.97 and RMSE of 1.3 cm [11]. No earlier study has reported on UAV-based height estimation of pumpkin plants but there was a study that compared the UAV-estimated and the ground truth measurements of pumpkin fruits height [32] with the mean fruit height of 12.4 cm.

However, our results showed a lower accuracy for barley and wheat, with $R^2$ values of 0.3563 and 0.4949, RMSE of 14.16 cm and 22.04 cm, and correlation coefficients for the growth curves of 0.32 and 0.52, respectively. For comparison, one study of a barley crop reported a higher correlation between UAV-estimated and ground truth plant height with $R^2$ of 0.92 and an underestimation of 10 cm [14]. The UAV plant height was underestimated because the plant height represents the mean plant height of all 0.01 m pixels in a plot, which resulted not only the top of the plant was measured but also the lower parts. A similar approach has been used in our study where buffers of 50 cm and 75 cm were applied to extract the average of UAV-estimated plant height. In our case, underestimations were also observed, especially during the early growth stages. The underestimation of plant height has also been reported for other cereal crops, such as wheat [21], maize [33], and sorghum [19].

The difference between cabbage and pumpkin on the one hand and barley and wheat on the other hand is that the former are crops with a closed canopy and the latter are crops with a very open canopy. In our intercropping field setup, the cereal crops were sown in a more open structure than that which is common in monocultures. Due to the open structure, a large part of the soil and of different wheats in the cereal strips were visible in the aerial images. This confused the SfM-MVS method in the reconstruction of the

DSM, which results in CHMs with severe underestimations of the height. Furthermore, we measured the highest tips of the cereal plants by hand, but because of the thin structures, the limited resolution of the camera, and the smoothing process in the SfM-MVS process, these tips were often ignored in the 3D reconstruction, which is a known issue [34]. On the other hand, during the later growth stage of the barley, the UAV-estimated plant heights were greater than the ground truth measurements. This is possibly due to the emergence of large weed plans covering the tops of the barley plants.

Our results suggest that height estimations based on UAV imagery are accurate for crops with a closed canopy, and less so for crops with an open canopy.

### 4.2. Limitation and Uncertainties That Affected Plant height Estimation

There are several factors that contributed to the limitation and uncertainties that can influence the UAV-based estimation of plant height for an intercropping production system. The first factor is the manual field data collection which introduced human errors in this study. The underestimation of barley and wheat crop for UAV plant height in the middle of the growing season (16 July 2020 and 24 July 2020) was caused by the wrong method of ground measurement. The spike of barley and wheat plants was bent up, which caused higher ground truth plant heights to be recorded. In this context, the method for the ground truth measurements in the field should be well decided according to the growth stages. Second, environmental factors also can potentially have an impact on the UAV-estimated plant height. For example, wind could cause large plant movements especially barley and wheat plant during UAV acquisition especially when the plants were taller. Moreover, wind also affects the stability of the UAV platform, which experienced vibration effects during acquisition. The unstable sensor positions results in image overlap inconsistency that influences the image matching [35]. These unfavorable conditions could affect the quality of a dense point cloud, reducing mesh quality, and thus causing higher error when DSMs are generated [36].

On the other hand, the UAV needs to capture large amounts of images to cover the small experimental field of view and low flight altitude to ensure high overlaps and optimal resolution for individual plants height estimation. However, this has resulted in increasing time for image matching, unstable aerial triangulation, and low elevation accuracy. We also faced some limitation due to the low flight altitude where the north part of the experimental field has larger and higher trees at the border of the intercropping field, which can affect the terrain interpolation. Clearing within the experimental field could adjust the interpolation process and obtain more points that could increase accuracy [37]. However, it will be impossible to remove the trees at the border of the experimental field; thus, a higher flight altitude should be used to reduce this problem.

Another factor that contributed to the limitations and uncertainties was the accuracy of the DSM, which strongly relates to the density of the point cloud generated by matching the feature points in RGB images using the SfM algorithm. A study [36] has shown the comparison of rapeseed height estimation accuracy using different point cloud qualities and treatments, where the construction of the crop canopy model was more detailed as the point cloud precision increased, and the inaccuracies in the point cloud generation process could complicate the SfM processing. In addition, the uncertainties of the DTM accuracy also influenced the CHM. This study used specific DTMs for each crop, as explained in Section 2.4.2. This approach may have introduced error, especially during the early growth stage. Normally, a DTM is obtained by conducting extra flight task either before or after growing season [33]. However, due to the intercropping setting, the field was often planted with cash crops which increases the difficulty of deriving the DTMs from the DSMs [38]. Another study reported obtaining data from the authorities database, but this approach is limited as it contained coarse spatial resolution [39]. On the other hand, the percentile selection also influenced the accuracy of the plant height. The used percentiles are commonly applied for study in cereal crops, such as barley and wheat. A specific percentile representing the pixel value on a plant height map is selected, such as the 100th

percentile [40], the 99.5th percentile [17], or the 50th percentile [41]. However, these values might not necessarily be the most suitable representative for plant height as the percentile shows that it is depending on the type of crop and the growth stage in this study (Figure 6).

### 4.3. UAV-Based Height Estimation of Individual Plants for the Study of Intercropping Systems

The overview of optimal settings for UAV-based crop height estimation, as presented in Table 4, may be a guideline to help growers and researchers to choose the appropriate settings to estimate individual plant height in intercropping fields for different crops. It is essential for the study of an intercropping field to have accurate height estimations of individual plants, for instance, for phenotyping research [42], to study phenotype–genotype relationships [43], or to make management decisions for growers throughout the whole growth period [44].

The ground truth measurement was conducted by 2 people and took approximately 10 h. Meanwhile, the acquisition by UAV took approximately 12 min and the creation of the canopy height models (CHM) with SfM-MVS was performed using Agisoft in approximately 17 h, requiring some manual inputs. The extraction of individual plant height from the CHM was performed in seconds using ArcGIS. Finally, the statistical analysis was performed with customized python scripts in a second. The most time consuming part is the SfM-MVS process, but this can easily be improved with parallel processing on multiple computers. In our experience, the availability of plant trait information within 24 h after UAV acquisition satisfies a large number of operational farming decision situations. For plant scientists, the duration of processing is even less critical. In the case of scenarios where UAV-based plant traits, such as height, are used for real-time management decisions, the use of an active technique, such as LiDAR, seems more appropriate, as this allows near real-time analysis of the measured point clouds from which crop height can be directly derived [45]. However, LiDAR-based UAV systems are still more expensive and require additional knowledge and experience on point cloud (pre-)processing.

### 4.4. Future Work

The results show that the height estimation for crops with a closed canopy, such as cabbage and pumpkin, can be performed reliably with UAV-based RGB images. However, for crops with an open canopy, such as barley and wheat, the CHM turned out not to not always be accurate, resulting in a low accuracy of the height estimations. Future work needs to focus on the SfM-MVS process to accurate CHMs for crops with an open canopy.

In the presented study, the plant locations in the orthomosaics were determined based on GPS measurements performed in the field. In a future application, the process of locating individual plants needs to be automated and performed in such a way that all plants in the field are localized. This can be performed by training deep neural networks to detect individual plants of multiple crop species, similar to the detection of individual trees in [27]. This will allow the monitoring of plant development over time of all plants in the field, providing a wealth of information for plant scientists and growers.

Future research will also need to include more crop species and a multitude of fields with different intercropping layouts, to test the general applicability of the methods.

### 5. Conclusions

In this study, the UAV-based height estimations of individual plants in an intercropping field were compared with ground truth measurements. After optimizing the buffer size and percentile, we can conclude that the height estimation for cabbage and pumpkin was more accurate than that for barley and wheat. The whole-season $R^2$ of individual plant heights for cabbage and pumpkin were 0.86 and 0.94, respectively, while those for barley and wheat were 0.36 and 0.49, respectively. Similarly, the whole-season RMSEs for cabbage and pumpkin were 6.75 cm and 7.00 cm, respectively, while those for barley and wheat were 14.16 cm and 22.04 cm, respectively. When comparing the estimated growth curves

per plant over the season with the ground truth, we observed average correlations for cabbage, pumpkin, barley, and wheat of 0.90, 0.96, 0.32, and 0.52, respectively.

There were several factors that influenced the results in this study, such as the weather conditions that influenced the quality of image matching and alignment, and the variability in ground truth sampling. The results suggest that UAV imagery can provide reliable height estimation of individual plants in an intercropping field for crops with a closed canopy, such as cabbage and pumpkin plants. However, the current approach cannot yet provide reliable plant height estimates for cereals with an open canopy, such as barley and wheat.

In future work, more advanced methods need to be developed to provide reliable height maps for crops with an open canopy. Furthermore, data should be collected over a longer period of time to cover more growth stages and for more crop species. Generally, the robustness of the method must be further investigated, as we only used data from a single experiment in one year. Processing of the data now still required some manual steps, which should be further automated in future work.

**Author Contributions:** Conceptualization, N.J., G.K. and L.K.; methodology, N.J., G.K. and L.K.; software, N.J., G.K. and L.K.; writing—original draft preparation, N.J.; writing—review and editing, N.J., G.K. and L.K. All authors have read and agreed to the published version of the manuscript.

**Funding:** This research was funded by the Young Academic Training Scheme Programme (SLAM), Universiti Malaysia Terengganu, Malaysia; UMT/PEND/500-4/4/2 JILID 3 (39).

**Institutional Review Board Statement:** Not applicable.

**Informed Consent Statement:** Not applicable.

**Data Availability Statement:** The data presented in this study are available upon request from the correspondence author at (norazlida.jamil@wur.nl).

**Acknowledgments:** The authors gratefully acknowledge colleagues Kevin Yao, Michiel Mans, Harm Bartholomeus, and Aldo Bergsma for their help in data collection and improvement of data analysis.

**Conflicts of Interest:** The authors declare no conflict of interest.

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
