# Peer review of "Evaluation of Individual Plant Growth Estimation in an Intercropping Field with UAV Imagery"

_agriculture, doi:10.3390/agriculture12010102_

Round 1

Reviewer 1 Report

The difference, importance and definition of the study are well done. There is also a detailed explanation.

1. However, wider information sharing is required for the conclusion part.

2. In addition, for the discussion section, the applicability of these applications with methods such as deep learning should be discussed instead of ready-made programming.

3. Also, discuss the efficiency of SFM for researchers interested in real-time applications. How long is the SFM processing time? What alternative methods can be used?

Reviewer 2 Report

The paper presents the original results of the estimation of individual plant height using UAV imagery. Performing research for an intercropping field is an innovative element. However, the usefulness of the applied procedure in practice is doubtful, especially for some plants. 

The authors emphasized the non-destructiveness of the technique. However, the authors did not indicate why non-destructiveness is an advantage. Are there destructive methods for estimating plant height? 

lines 105-106: What plants were grown in 2018 and 2019? 

line 109: Organic standards should be described in more detail.

Please explain all abbreviations used in Figure 1b.

lines 111, 114, 135: What does "Field 3" mean?  

Has the experiment been carried out only in this Field 3 or also in others?

Table 1 (first column - number of plants): Please specify why such numbers of measurements have been determined. Please justify the size of the samples. 

Table 1: Can the results depend on flight time and illumination? This is not discussed in detail. The discussion includes mainly information about the influence of wind.

lines 169-170: It has been written: "The total hours for whole field to do manual measurements was approximately 10 hours for all four selected crops depending on the weather conditions." How does this time refer to the time of the analysis using UAV Imagery including image acquisition, image processing and statistical analysis? 

line 170: Why have four crops been selected?

line 392: It has been written: "wrong ground truth measurement has resulted in low correlation coefficient". I think that one season is not enough for this type of experiment. Research should be repeated in the second season performing all measurements correctly.

Most of the results presented in Figure 9 are not satisfactory. Does each curve include a mean value for two plants with the highest and lowest correlation coefficient values? Is there another way to present the results? 

I think that further studies for other seasons and/or other locations are necessary. Authors should present the results of validation of the applied procedure.

Round 2

Reviewer 2 Report

The manuscript has been corrected according to the comments. I recommend accepting the present form of the manuscript.